# The Impact of a Congested Match Schedule (Due to the COVID-19 Lockdown) on Creatine Kinase (CK) in Elite Football Players Using GPS Tracking Technology

**DOI:** 10.3390/s24216917

**Published:** 2024-10-28

**Authors:** Jorge Garcia-Unanue, Antonio Hernandez-Martin, David Viejo-Romero, Samuel Manzano-Carrasco, Javier Sanchez-Sanchez, Leonor Gallardo, Jose Luis Felipe

**Affiliations:** 1IGOID Research Group, Physical Activity and Sport Sciences Department, University of Castilla-La Mancha, 45071 Toledo, Spain; leonor.gallardo@uclm.es (L.G.); joseluis.felipe@uclm.es (J.L.F.); 2School of Sport Sciences, Universidad Europea de Madrid, Villaviciosa de Odón, 28670 Madrid, Spain; antonio.hernandez@universidadeuropea.es (A.H.-M.); david.viejo@universidadeuropea.es (D.V.-R.); javier.sanchez2@universidadeuropea.es (J.S.-S.); 3Department of Communication and Education, Universidad Loyola Andalucia, 41704 Sevilla, Spain; smanzano@uloyola.es; 4Performance Analysis Department, UD Las Palmas, 35019 Las Palmas de Gran Canaria, Spain

**Keywords:** creatine kinase, external load, fatigue, injury risk, physical demands

## Abstract

The aim was to analyse the consequences of a congested schedule (due to the COVID-19 lockdown) on creatine kinase (CK) in elite football players using GPS tracking technology. A total of 17 elite football players were monitored in training and competition with a global positioning system. Variables including total distance, high-intensity distance, and distance acceleration and deceleration were analysed. Different measurements of serum creatine kinase (CK) concentration were performed on match day (MD) and at 24 h (MD+1), 48 h (MD+2), and 72 h (MD+3) after each match to study the muscle damage of each individual player caused during the match. The results showed a significant increase in physical demands in training (in relative terms regarding the match) at MD+3 compared to MD+1 and MD+2. Furthermore, at +72 h, CK decreases to a value almost the same as that before MD. On the other hand, the players with lower demands for high-intensity actions in the match showed a higher reduction in the concentration of CK at MD+3 compared to MD+1 and MD+2 (*p* < 0.05). It became evident that players with high-intensity demand and a high number of accelerations and decelerations need more time to assimilate the match load and can remain in a state of muscle fatigue for up to 3 days. In addition, a congested schedule can lead to a state of chronic fatigue in elite football players, limiting physical performance and possibly increasing the potential risk of injury for football players.

## 1. Introduction

Football stands out because of the intermittent physical participation of the players according to their position and the moment of the game, which implies periods in which the players perform high-intensity activity interspersed with actions of lower intensity or recovery [1]. During the 90 min match, each player covers an average total distance of approximately 11 km, with 5% of this distance covered at high speed and 3% at sprint speed [2]. The ability to perform and repeat these intense actions during the match has been considered one of the key factors for the performance of football players [3]. It has also been shown that this characteristic is evolving in football such that high-intensity demands have increased significantly, with an increase in sprint distance of approximately 35% over a period of seven seasons [4]. In addition to the activity carried out at high speed, players perform between 1000 and 1400 actions of high intensity and short duration, ranging between 3 and 5 s, and involve actions with and without the ball that alternate randomly [5]. These efforts require a good number of eccentric muscle contractions, which greatly contribute to the muscular stress suffered by the player and are perceptible up to 120 h after the match [6].

In this sense, monitoring systems have become increasingly important to assess these variables. Among the main monitoring methods, the global positioning system (GPS) is widely used in elite football to quantify training and competition load [7]. GPS is one of the current models of external load control to monitor the movement patterns and physical actions performed by football players during training and matches [8], as well as to help players avoid injuries [9].

The competitive demands of football involve various physiological systems, including the musculoskeletal, nervous, immune, and metabolic systems, to the point that recovery strategies after exercise influence the players’ preparation for the next match [10]. Different studies have been carried out with the objective of analyzing this fatigue and recovery, including the analysis-specific variables of performance such as biochemical markers and muscle status, to achieve greater efficiency of progress during recovery [11].

Therefore, these parameters of muscle metabolism, including creatine kinase (CK), lactate dehydrogenase (LDH), and myoglobin, tend to increase after exercise [12]. CK has been used as an indirect marker of muscle damage to quantify and determine the extent of muscle damage caused during competition and training [13]. After a high-intensity effort, the main function of CK in muscle is to provide phosphorus for adenosine triphosphate resynthesis [14]. This parameter increases after exercise, and it peaks between 24 and 48 h post match [15]. Therefore, CK is measured between this period of time after the match when there is no decrease in serum concentration (48 h after), while the next assessment is carried out between 48 and 72 h to determine if the athlete recovers for the next match [16]. An insufficient recovery can adversely affect physical performance [17].

This is an increasingly important aspect in modern football, as it involves a large number of matches during the season, and it is not unusual for a team to play two matches in a week [18]. A congestion of matches can lead to a lack of motivation and concentration, which can affect coordination [19]. Because of low recovery between matches, residual fatigue occurs [20] and increases the stress imposed on the players, which decreases performance [21]. It is necessary to know the impact and physiological changes induced by a football match to help design and develop more effective strategies to shorten the duration to a full recovery [22]. In this regard, further studies are required to demonstrate the impact of high-level match congestion on the muscle damage response of professional players.

Due to the COVID-19 pandemic, there has been a congested fixture period calendar after the resumption of competition in Spain in May 2020. This has influenced the physical performance and injury rate after the quarantine period [23]. Therefore, the quantification of CK and physical demands using GPS devices is ideal to analyse physical performance and muscle damage during the congestion period of training and matches. Therefore, the aim of this study was to analyse the influence of a congested match schedule on physical performance and muscle damage in professional football players after the COVID-19 lockdown.

## 2. Materials and Methods

### 2.1. Sample

A longitudinal and quantitative study was performed with male professional football players who played in the Spanish 2nd Division (LaLiga Smartbank) during the 2019–2020 season. All the information was collected, and the data were obtained during the last phase of the LaLiga SmartBank. This period coincides with the resumption of competition after the lockdown period in Spain. Specifically, this period was between 12 June 2020 and 20 July 2020. All the players played 11 matches and participated in 23 training sessions. The study protocol was approved and followed the guidelines established by the local institution, the Ethics Committee of the European University of Madrid (CIPI35/2020), and it was in accordance with the recommendations of the Declaration of Helsinki. The players were previously informed through a document about the purpose of the study and the nature of the tests that would be performed, and an informed consent form was signed prior to the tests.

### 2.2. Procedures

The data from each player in each match were considered as one observation. Only data of players who participated for at least 10 min in the match and had a complete measurement of physical and physiological variables were included in the study. Furthermore, goalkeepers and players who were penalised or injured were excluded from the analysis and sampling, as were football players not participating in the matches. The players were evaluated individually once before a training session and match competition.

The final study consisted of 17 male professional football players (25.91 ± 3.13 years; 71.27 ± 3.25 kg; 179.36 ± 5.14 cm) divided into three subsamples according to the demand for each of the physical variables analysed (total distance, high-intensity distance [distance travelled above 21 km/h], high-intensity acceleration distance [acceleration distance travelled above 3 m/s^2^], and high-intensity deceleration distance [deceleration distance travelled above −3 m/s^2^]). The three final subgroups consisted of low-physical-demand players (LPD), medium-physical-demand players (MPD), and high-physical-demand players (HPD).

### 2.3. Equipment

All team players wore inertial measurement devices (IMUs) (81 × 45 × 16 Mm; 65 gr) with 18 Hz GPS tracking technology (WIMU PRO™, Almería, Spain) to evaluate their movement patterns during each match and training session. For the analysis and data extraction, the software SPROTM v. 960 (REALTRACK SYSTEMS S.L., Almería, Spain) was used. The precision and reliability of this GPS system have been reported in previous investigations [24].

CK measurements were performed at four time points: immediately at the end of the game (MD) as a baseline measure, 24 h after the game (MD+1), 48 h after the game (MD+2), and 72 h after the game (MD+3). Blood samples were collected from the index finger using test strips (REFLOTRON TEST STRIPS^®^, Roche, Switzerland) and measured on a biochemical analyser (REFLOTRON PLUS^®^, Roche, Switzerland). The doctor responsible for the medical service was in charge of taking the measurements. Measurements were made according to the established hours, with a variation of ±1 h. Before each sampling, the instrument was calibrated according to the manufacturer’s recommendations. A CK measurement protocol was developed according to previous studies [25]. CK sample collection was carried out in the doctor’s office. This office was perfectly equipped and prepared, with a standard temperature of 21 °C and a relative humidity of 65%, fulfilling conditions of suitability and homogeneity for carrying out the measurements.

### 2.4. Statistical Analysis

Results are presented as means ± standard deviations. The three subsamples based on the physical variables (i.e., LPD, MPD, and HPD) were created using k-means clustering. The Kolmogorov–Smirnov test revealed a non-normal behaviour of all variables; therefore, nonparametric tests were performed. The Kruskal–Wallis H test was used to compare the physical parameters and CK blood levels in % relative to the match between MD+1, MD+2, and MD+3. The same test was used to compare the CK blood levels in % relative to the match between MD+1, MD+2, and MD+3 in each of the three subsamples of each physical variable, and the CK blood levels in absolute terms and in % relative to the match between the three subsamples of each physical variable at each of the time points (i.e., MD, MD+1, MD+2, and MD+3). Here, differences were identified, and Dunn–Bonferroni tests were performed for post hoc pairwise comparisons. SPSS V24.0 for Windows (SPSS Inc., Chicago, IL, USA) was used for all tests. The level of significance was set at *p* < 0.05

## 3. Results

The analysis of variance revealed significant differences in the physical performance and muscle damage of the players in the days after the match compared to the results obtained during the MD (Figure 1; *p* < 0.001). The external load of the players at MD+3 showed a significant increase compared to MD+1 and MD+2 for all the variables analysed (*p* < 0.001; ES: 0.27–0.96). The CK results showed a significant reduction at MD+3 compared to MD+1 (−202.32%; ES: 0.70; *p* < 0.001) and MD+2 (−187.46%; ES: 0.64; *p* < 0.001).

The subgroup analysis revealed a significant influence of the physical demands of the match on the evolution of the CK levels of the players (Table 1; *p* < 0.001). In relation to the total distance covered, the players with the greatest distances covered in the match (HPD) showed a significant reduction at MD+3 compared to MD+1 (−244.37%; ES:0.78) and MD+2 (−191.15%; ES: 0.66) according to the baseline situation (*p* < 0.05). In this sense, in relation to the distance at high intensity and the distance in high-intensity acceleration, the groups with LPD and MPD revealed a reduction in the concentration of CK at MD+3 compared to MD+1 and MD+2 (*p* < 0.05). Finally, the subgroup analysis regarding the distance covered during high-intensity deceleration showed a significant decrease at MD+3 compared to MD+1 in the three groups analysed (*p* < 0.05).

The analysis of the three subgroups only revealed a greater concentration of CK in the MPD group compared to the HPD group (+176.84%; ES: 0.43) as a function of the distance covered during high-intensity acceleration at MD+1 (*p* < 0.05). In absolute terms, the comparative analysis showed higher levels of CK in the blood in the HPD group before the game as a function of the distance travelled at high intensity (+43.66 U l^−1^; ES: 0.54) and during high-intensity accelerations (+64.24 U l^−1^; ES: 1.03) compared to the LPD group (*p* < 0.05). In relation to the results obtained 24 h after the match, the MPD group revealed higher blood concentrations of CK than the LPD group (+166.76 U l^−1^; ES: 1.24) as a function of the distance covered during high-intensity accelerations (*p* < 0.05).

## 4. Discussion

The aim of this study was to analyse the influence of a congested match schedule on physical performance and muscle damage in Spanish professional football players after the COVID-19 lockdown. For this fatigue assessment, internal load (serum CK measurements) and external load (GPS variables) were used individually. Our findings demonstrate that players with higher-intensity physical demands in matches, recognized as distance covered at high intensity and distance covered accelerating and decelerating at high intensity, had greater muscle damage and required a longer recovery time. Also, large differences were observed in the recovery time, with 72 h being the ideal time. A long period of inactivity due to the COVID-19 lockdown caused significant negative adaptations in the players, which are related to increases in fitness-dependent injuries [26]. Therefore, a long preseason is of crucial importance because of its protective effect, as it reduces injury risk and injury severity and increases player availability during the season [27]. In this sense, preparing for 4 weeks with limiting guidelines on the return to competition is a handicap to achieve adequate adaptations for competition.

After an analysis of the chronological evolution of the concentration of CK in plasma, an increase in the curve of this biochemical marker is observed from immediately after the end of the match up to 4 days after the end of the match [18]. The highest CK concentration occurred in the 24 h post-match test [28]. In this study, the peak CK levels differ and are lower than reference levels in other studies. This important finding could be due to the players remaining in a state of chronic fatigue for much of this period, which prevented the logical manifestation of this biomarker. Indicators of muscle damage have been shown to decrease as a result of frequent bursts of eccentric loading, or as a result of continued sporting activity [29], as is the case with a congested schedule in football. Therefore, high loads applied over a long period of time by football players can lead to metabolic and organic overloads and, consequently, to chronic fatigue [30].

The release of CK in plasma and its elimination by the body depend on the level of training of the subject and the type, intensity, and duration of exercise [31]. Our results are focused on the characteristics of the effort, which were analysed individually. After 48 h post match, the CK level was still high [15,17]. This suggests that the players did not have a complete recovery 48 h after the match. However, at 72 h, there was a significant reduction in serum CK concentrations compared to previous tests (MD+1 and MD+2). This decrease was evidenced previously [31], where the CK levels decreased in the days after the match, probably as an adaptation to the training stimuli that produced the muscle damage. Therefore, after 72 h, minimal functional recovery and reduced risk of injury can be guaranteed [24]. In this way, physical demands and neuromuscular function show a reduction 24 h and 48 h post match. This decrease in training performance is explained by the structured planning due to the congested match period, and thus the loads are adjusted on the day of the match [20].

On the other hand, a large requirement of the physical demands of the players in the competition influence the concentration of CK and the muscle damage of the players in subsequent days. Changes in the CK levels after the match have higher significant correlations with high-speed running than total distance covered [13]. Our findings indicated a positive relationship between the total distance covered and the time of recovery, where the players with the greatest distance covered in the competition showed a significant reduction at MD+3 compared to groups with less load. This did not happen in other studies that did not find relationships between total distance and the increase in CK concentration in football.18 In addition, other studies have shown a negative correlation between total distance and CK levels at MD+1 and MD+2 [28]. Therefore, players who are stronger and have more strength in the lower body show reduced levels of CK 48 h after the match [32]. This implies that the muscle damage produced by travelling a certain distance does not generate excessive fatigue, which is quantified using concentrations of CK. This may be because players with more fatigue resistance can travel long distances in matches due to the predominance of muscle fibres I or IIb, which are associated with less muscle damage than type II muscle fibres [19].

Otherwise, players who run long distances at high speeds develop increased muscle damage, as demonstrated by post-match CK measurements [17]. These findings are similar to our results, which show significant reductions in the concentrations of CK in the groups with low and medium demand regarding these variables, with respect to the group with HPD in competition. This decrease is especially significant in the MD+3 test, which uses the MD+1 and MD+2 measurements as references. On the other hand, another study showed that high-intensity match activities are related to CK levels at 24 h after the football match [7].

The main take-home application of this study is that it is necessary to adapt the training regimen during the three days after a competition by considering the high load experienced by the players as well as the total volume of minutes played. These players have a high state of fatigue, and their demands during post-match training have been shown to be lower. In addition, it is important that the time saved by the low load of these workouts is used to apply recovery measures. Thus, if there are less than 72 h between competitions, the degree of recovery should be assessed in order to make the lineup and design the match plan. It is possible to save the load, and the performance of these players during the competition is not limited, which takes into account the demands of the previous match.

There are some limitations to be considered when interpreting the present results. Firstly, we do not have data prior to the lockdown to be able to make a pre–post comparison. This is due to the unexpected cancellation of the competition due to COVID-19. Moreover, dietary intake was not monitored and could influence recovery time. Furthermore, factors such as the quality of sleep and the players’ activity outside of training hours were also not analyzed.

## 5. Conclusions

Once the physical results obtained with the GPS devices and the CK values are known, it is essential to manage the workload of players during training with a congested match schedule to ensure adequate recovery for the next match. Special attention should be paid in the days after the match to players who engaged in higher-intensity efforts and high acceleration and deceleration activities during the match. These players will need more time to assimilate the match loads and will not be in an acceptable state of recovery until 3 days after the match. Therefore, with a congested match schedule, a rotation of players is essential for recovery and for the players to be able to perform at an optimal physical level in matches. Organisers of elite tournaments, such as leagues or national team competitions, must be aware of the risks faced by elite players during congested calendar situations and opt for the health and well-being of the players over the spectacle.

## Figures and Tables

**Figure 1 sensors-24-06917-f001:**
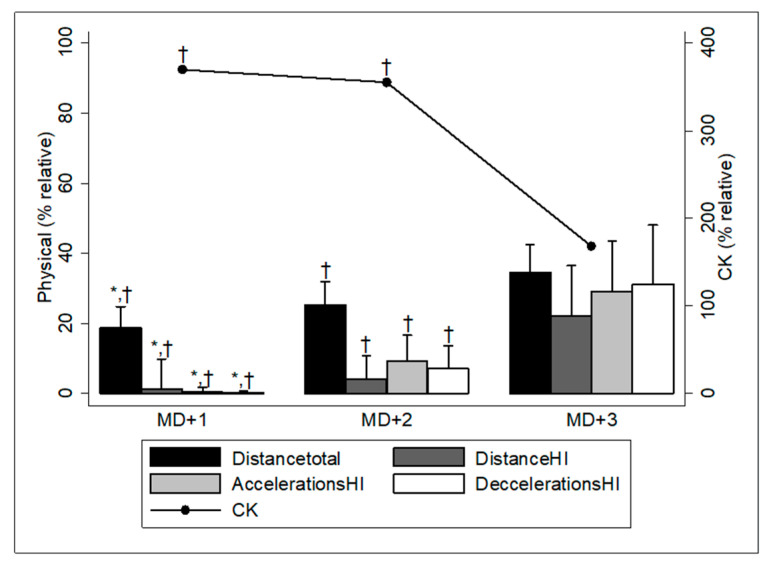
Physical demands and CK blood levels at MD+1 (24 h), MD+2 (48 h), and MD+3 (72 h) according to the MD results. HI: high intensity; CK: creatine kinase. * Significant differences with respect to 48 h; ^†^ significant differences with respect to 72 h.

**Table 1 sensors-24-06917-t001:** CK blood levels at MD, MD+1, MD+2, and MD+3 in professional football players according to the physical demands.

Subgroups		Mean			CK MD			CK MD+1			% MD+1			CK MD+2			% MD+2			CK MD+3			% MD+3		
**Total distance (m)**	LPD	2744.33	±	1281.18	51.28	±	59.05	149.80	±	67.36	619.44	±	627.35	118.83	±	124.55	388.24	±	541.60	17.00	±	9.90	−56.13	±	46.40
	MPD	6629.50	±	1166.34	96.78	±	77.40	176.80	±	111.56	179.92	±	177.63	132.16	±	103.37	211.49	±	371.54	63.07	±	53.39	21.49	±	199.72
	HPD	9692.32	±	769.53	89.08	±	73.70	250.10	±	214.21	310.34	±	414.56 ^c^	207.12	±	185.08	257.12	±	371.01 ^c^	94.73	±	84.64	65.97	±	208.53 ^a,b^
**High-intensity distance (m)**	LPD	214.54	±	101.43	74.08	±	97.18 *	141.71	±	89.92	355.61	±	428.16 ^c^	132.29	±	126.57	280.31	±	420.03 ^c^	59.15	±	68.52	13.08	±	182.71 ^a,b^
	MPD	470.82	±	74.94	81.63	±	63.71	261.33	±	222.65	326.29	±	396.26 ^c^	202.50	±	186.38	291.06	±	406.10 ^c^	82.91	±	71.47	68.57	±	225.82 ^a,b^
	HPD	785.53	±	112.38	117.74	±	63.48 ^¥^	247.49	±	187.94	251.74	±	472.24	214.77	±	175.84	154.78	±	248.26	126.75	±	104.24	54.34	±	157.48
**High-intensity acceleration distance (m)**	LPD	309.00	±	102.73	54.39	±	55.01 *	120.12	±	61.48 ^#^	358.32	±	391.37 ^c^	115.85	±	92.70	369.37	±	504.84 ^c^	50.35	±	47.87	67.45	±	275.72 ^a,b^
	MPD	557.54	±	59.86	88.48	±	79.38	286.88	±	206.80 ^¥^	380.93	±	452.73 ^c^*	222.04	±	197.78	263.45	±	347.54 ^c^	104.41	±	96.67	62.92	±	195.78 ^a,b^
	HPD	774.93	±	83.77	118.63	±	69.24 ^¥^	248.30	±	222.19	204.09	±	379.59 ^#^	218.47	±	183.99	150.81	±	246.22	99.93	±	67.52	20.67	±	84.65
**High-intensity deceleration distance (m)**	LPD	103.55	±	61.48	69.37	±	71.06	155.23	±	75.13	550.61	±	526.00 ^c^	152.94	±	124.41	385.97	±	462.35 ^c^	55.00	±	56.36	−33.09	±	31.93 ^a,b^
	MPD	304.47	±	46.09	87.42	±	75.03	240.50	±	227.57	215.20	±	264.39 ^c^	185.83	±	189.00	222.23	±	356.59 ^c^	84.38	±	75.35	65.65	±	239.01 ^a,b^
	HPD	443.74	±	48.30	97.38	±	72.05	242.64	±	158.66	426.66	±	554.48 ^c^	215.55	±	156.52	290.23	±	402.67	103.99	±	96.34	64.31	±	151.04 ^a^

^a,b,c^ Significant differences between MD+1, MD+2, and MD+3, respectively; * Significant differences with HPD group; ^#^ Significant differences with MPD group; ^¥^ Significant differences with LPD group; MPD: match day; CK: creatin kinase; LPD: low physical demand players; MPD: medium physical demand players; HPD: high physical demand players.

## Data Availability

Data that support the findings of this study are available on request from the corresponding author. The data are not publicly available due to contain information that could compromise the privacy of the participants.

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
