# Peer review of "The Impact of a Congested Match Schedule (Due to the COVID-19 Lockdown) on Creatine Kinase (CK) in Elite Football Players Using GPS Tracking Technology"

_sensors, 2024, doi:10.3390/s24216917_

Round 1
Reviewer 1 Report
Comments and Suggestions for Authors
The submitted manuscript for evaluation is an interesting research project. The combination of the external load indicator, which are the indicators recorded by the GPS system, with the internal load indicator - CK, has not been studied so far. The term COVID19 appears in the title of the paper. The timing of the study after lockdown is very specific. During the lockdown period, there was a suspension of league play. There was also a reduction in training activity. One would therefore expect a comparative compilation of the characteristics of the relationship between external and internal load before lockdown and after the start of post-lockdown games. Of course, no one planned lockdown as a variable that modifies training. That is, it would be necessary to juxtapose the correlations of the period immediately after the lockdown with the values, for example, 6-8 months later. Then it should be assumed that the level of preparation would no longer be determined by the break in training and competition. This is the first comment on the organization of the study, The second is the timing of blood collection and CK determinations. A post-exercise blood draw, even directly, cannot be considered to determine the baseline level. Subsequent measurements at 24, 48 and 72 hours after the match are subject to the values of CK changes that occurred as a result of training activities between matches.
In addition to these two comments relating to the organization of the study and thus the quality of the material collected, the following points should be noted:
- the lack of CK profiles of athletes from before the experimental period. CK is a highly individualized indicator and reference values are highly variable. With individual profiles, it would be possible to operate the difference and thus standardize within the group the range of CK changes.
- No differentiation of external loads into match and training loads
- There is no justification for the inclusion in the study group of a player who played in the match for 10 minutes. Here the difference in working time between 10 minutes and 60 or 90 minutes is highly significant for CK values;
- explain the %Δ index, i.e. the formula for determining once the essence of interpreting the value. This is especially important when the standard deviation exceeds the mean value.
- it is difficult to find an interpretation for CK changes without specific amounts of training loads in the 72 hours after the match.
- strength training stimulates muscle damage and CK increase, whether it was applied at 72 hours after the match;
- no clear evidence of a relationship between the magnitude of the internal load and changes in CK values
- conclusion is very general. It does not add any new information to the topic of the manuscript. This section is unacceptable in its entirety.
The manuscript needs very extensive editorial revisions. The possibility of more readable illustrations and tables should also be studied. The current level of the text of the work does not allow to express a positive opinion.-
Author Response
The submitted manuscript for evaluation is an interesting research project. The combination of the external load indicator, which are the indicators recorded by the GPS system, with the internal load indicator - CK, has not been studied so far. The term COVID19 appears in the title of the paper. The timing of the study after lockdown is very specific. During the lockdown period, there was a suspension of league play. There was also a reduction in training activity. One would therefore expect a comparative compilation of the characteristics of the relationship between external and internal load before lockdown and after the start of post-lockdown games. Of course, no one planned lockdown as a variable that modifies training. That is, it would be necessary to juxtapose the correlations of the period immediately after the lockdown with the values, for example, 6-8 months later. Then it should be assumed that the level of preparation would no longer be determined by the break in training and competition. This is the first comment on the organization of the study, The second is the timing of blood collection and CK determinations. A post-exercise blood draw, even directly, cannot be considered to determine the baseline level. Subsequent measurements at 24, 48 and 72 hours after the match are subject to the values of CK changes that occurred as a result of training activities between matches.
Dear Reviewer, thank you very much for your comments and insights. They are very valuable for improving the quality of the manuscript. This study was not designed as a laboratory study but as a real-world environment study. Therefore, after the lockdown was declared in Spain, the players stayed in their homes for a month.
Afterward, training and competition resumed, with 11 matches played over 4 weeks (a congested schedule). As a result, we did not have baseline data for the subjects, and blood collection could only be done immediately after each match, and at 24, 48, and 72 hours post-competition (considering that with a congested schedule of every 72 hours or less, the training sessions were regenerative).
The authors do not intend this to be a laboratory study, but rather to show the reality of what happens to elite footballers when subjected to a congested schedule. We have tried to control all extraneous variables, but we are aware of the study's limitations. Therefore, we were unable to implement multiple groups (control and experimental) or conduct a more controlled study. This is a real football team demonstrating the reality of what occurs in the day-to-day life of a football team.
In addition to these two comments relating to the organization of the study and thus the quality of the material collected, the following points should be noted:
- the lack of CK profiles of athletes from before the experimental period. CK is a highly individualized indicator and reference values are highly variable. With individual profiles, it would be possible to operate the difference and thus standardize within the group the range of CK changes.
Indeed, in practice, we used the data to individualize the match load (i.e., the minutes each player played based on the recorded CK values). However, as a limitation of the study, we could not access baseline data prior to the experimental period. In a situation where there is one match per week, we do not record CK because the player clears CK levels after 72 hours, even if the training load is high. We only record CK data when there is an unexpected situation, such as the resumption of the league after COVID-19. Therefore, we do not have this prior information.
- No differentiation of external loads into match and training loads.
Due to the situation of 11 matches in 4 weeks, the training sessions were 100% regenerative, as often less than 72 hours passed between matches.
- There is no justification for the inclusion in the study group of a player who played in the match for 10 minutes. Here the difference in working time between 10 minutes and 60 or 90 minutes is highly significant for CK values;
This was not for the study; we are demonstrating the reality of the method we use to monitor the CK levels of the team in a real competition scenario. Therefore, we could not divide the players into more precise groups. As we mentioned earlier, we are trying to showcase the actual game situation. This is not an experimental study.
- explain the %Δ index, i.e. the formula for determining once the essence of interpreting the value. This is especially important when the standard deviation exceeds the mean value.
We change this value. %Δ is the same that %relative respect to the match, the same indicator that is used in figure 1. Therefore, we change %Δ to have uniformity in the manuscript.
- it is difficult to find an interpretation for CK changes without specific amounts of training loads in the 72 hours after the match.
There is no training intervention. Only 100% regenerative training sessions (MD+1 and MD+2).
- strength training stimulates muscle damage and CK increase, whether it was applied at 72 hours after the match;
There was no strength training, only regenerative training sessions.
- no clear evidence of a relationship between the magnitude of the internal load and changes in CK values
We tried to distribute the playing minutes proportionally to avoid significant CK deviations. Therefore, all players have a similar training and competition load.
- conclusion is very general. It does not add any new information to the topic of the manuscript. This section is unacceptable in its entirety.
We have rewritten some parts of the conclusions, and we believe they are now more specific.
The manuscript needs very extensive editorial revisions. The possibility of more readable illustrations and tables should also be studied. The current level of the text of the work does not allow to express a positive opinion.
Reviewer 2 Report
Comments and Suggestions for Authors
General comments: Well conducted and written study. Regarding formal aspects, the manuscript is well structured and contains the following sections: Introduction; Materials and Methods; Results; Discussion ending with Conclusions (which must be reconsidered, the conclusions are not consistent with the purpose of the study and what was actually tested). The title could be improved. The language is appropriate, the text is clear, precise and objective. There is no evidence of plagiarism in the manuscript.
I hope these suggestions and comments are helpful!
Title: It's very big; I suggest writing CK in full; biochemical analysis is redundant, to analyze CK, it is logical to perform a biochemical analysis; and “performance after covid-19 lockdown” It suggests and induces us to hope that we will be able to identify one before and after the period of lockdown.
Just Ideia:
> The impact of a congested match schedule (by covid-19 lockdown) on creatine kinase (CK) in elite football players using GPS tracking technology.
>The impact of a congested match schedule (after covid-19 lockdown) on creatine kinase (CK) in elite football players using GPS tracking technology.
Abstract: It is appropriate (objective, methods, main results, and conclusion) and reflects the content of the manuscript.
L16 – The aim was … a congested schedule on [creatine kinase (CK)] [in] …
L30 - Think if it doesn't seem like speculation because you didn't test for injuries.
Introduction:
L41 – in modern football … (reference [4] is from 2014)
L57 – in recent years… (reference [11] is from 2008, It's not so new.)
L64-65 – Reference 14 does not seem to be appropriate to me, please check it
L66-67 – The same for reference 15, does not seem to be appropriate. They tested diastolic and systolic blood pressure, heart rate, lactate, testosterone and cortisol concentrations? Please check it!
L72-74 – check reference 18, it does not looks support the statement
L74-75 – the same problem check reference 19
L76-77 – the same for reference 21, check it, they do not investigated risk of injury.
Materials and Methods: The methodological procedures are appropriate to the study of the research question and are sufficiently detailed and described in an appropriate, clear, and objective manner. Here are a few things to consider:
L128 – Is the reference number 24 correct, or has it been changed? The study cited does not use GPS.
L144 - Is it appropriate to report means and SDs if you tested for non-normal behaviour for all variables? Just as you used non-parametric tests for comparison, for measures of central tendency, Medians and interquartile range (IQR) should be presented (Field, A. (2024). Discovering Statistics Using IBM SPSS Statistics. Reino Unido: SAGE Publications.).
Results: Presents the results in an appropriate manner, highlighting the main findings and avoiding unnecessary repetition. Use tables and figures as appropriate and facilitate appropriate dissemination of results. I have a few suggestions for you
L165 - in the caption of figure 1: I suggest putting the time after MD+ in parentheses, for example: MD+1 (24h), MD+2(48h) … This facilitates the reader and possible autonomy of your figure, for example in a lecture that uses only the figure without supporting text.
Discussion: The discussion is well led and helps to reflect from the results. I have some suggestions:
L226-227- irrelevant to the discussion session, it is of methods
L297- Devices and biochemical tests were not tested and analyzed in this study to draw conclusions about the validity of either.
Conclusions:
L306-310 - This conclusion is not consistent with the purpose of the study. Devices and biochemical tests were not tested and analyzed in this study to draw conclusions about the validity of either.
Author Response
General comments: Well conducted and written study. Regarding formal aspects, the manuscript is well structured and contains the following sections: Introduction; Materials and Methods; Results; Discussion ending with Conclusions (which must be reconsidered, the conclusions are not consistent with the purpose of the study and what was actually tested). The title could be improved. The language is appropriate, the text is clear, precise and objective. There is no evidence of plagiarism in the manuscript.
I hope these suggestions and comments are helpful!
Title: It's very big; I suggest writing CK in full; biochemical analysis is redundant, to analyze CK, it is logical to perform a biochemical analysis; and “performance after covid-19 lockdown” It suggests and induces us to hope that we will be able to identify one before and after the period of lockdown.
Just Ideia:
> The impact of a congested match schedule (by covid-19 lockdown) on creatine kinase (CK) in elite football players using GPS tracking technology.
>The impact of a congested match schedule (after covid-19 lockdown) on creatine kinase (CK) in elite football players using GPS tracking technology.
Thank you very much for your comment. The title has been changed based on its proposals.
Abstract: It is appropriate (objective, methods, main results, and conclusion) and reflects the content of the manuscript.
L16 àThe aim was … a congested schedule on [creatine kinase (CK)] [in] …
Changed.
L30 à Think if it doesn't seem like speculation because you didn't test for injuries.
Thank you for your suggestion. The sentence has been changed.
Introduction:
L41 à in modern football … (reference [4] is from 2014)
The sentence has been modified by deleting "modern".
L57 à in recent years… (reference [11] is from 2008, It's not so new.)
The sentence has been changed.
L64-65 à Reference 14 does not seem to be appropriate to me, please check it
Thank you very much for your comment. The reference has been changed.
L66-67 à The same for reference 15, does not seem to be appropriate. They tested diastolic and systolic blood pressure, heart rate, lactate, testosterone and cortisol concentrations? Please check it!
Thank you very much for your comment. The reference has been changed.
L72-74 à check reference 18, it does not looks support the statement
Thank you again. The reference has been checked and has been changed.
L74-75 à the same problem check reference 19
The reference has been changed.
L76-77 à the same for reference 21, check it, they do not investigated risk of injury.
The sentence has been changed
Materials and Methods: The methodological procedures are appropriate to the study of the research question and are sufficiently detailed and described in an appropriate, clear, and objective manner. Here are a few things to consider:
L128 à Is the reference number 24 correct, or has it been changed? The study cited does not use GPS.
Thank you for your comment. The reference has been checked and has been changed.
L144 - Is it appropriate to report means and SDs if you tested for non-normal behaviour for all variables? Just as you used non-parametric tests for comparison, for measures of central tendency, Medians and interquartile range (IQR) should be presented (Field, A. (2024). Discovering Statistics Using IBM SPSS Statistics. Reino Unido: SAGE Publications.).
The studies in this field present means because is easier to compare and understand in the sector, even if they are non-parametric variables. Therefore, the same dynamic has been followed with the aim of facilitating the reach of the article in the area.
Results: Presents the results in an appropriate manner, highlighting the main findings and avoiding unnecessary repetition. Use tables and figures as appropriate and facilitate appropriate dissemination of results. I have a few suggestions for you:
L165 à in the caption of figure 1: I suggest putting the time after MD+ in parentheses, for example: MD+1 (24h), MD+2(48h) … This facilitates the reader and possible autonomy of your figure, for example in a lecture that uses only the figure without supporting text.
Thank you very much for your suggestion. The time has been added to make it clearer for the reader.
Discussion: The discussion is well led and helps to reflect from the results. I have some suggestions:
L226-227- irrelevant to the discussion session, it is of methods
Thank you very much for the suggestion, the sentence has been removed.
L297- Devices and biochemical tests were not tested and analyzed in this study to draw conclusions about the validity of either.
Thank you very much for the suggestion. The sentence has been removed.
Conclusions:
L306-310 - This conclusion is not consistent with the purpose of the study. Devices and biochemical tests were not tested and analyzed in this study to draw conclusions about the validity of either.
Thank you very much for your comment, the conclusion has been modified.
Round 2
Reviewer 1 Report
Comments and Suggestions for Authors
None of my comments caused any reflection on the part of the authors of the manuscript. I believe that disqualifying errors in the work have not been corrected. Thus, the final evaluation is negative. Research carried out in training practice cannot be deprived of methodological correctness. With such an analysis of the collected material as in the manuscript, it is difficult to call the effect of the analysis and conclusions reliable.
Author Response
# Reviewer
The submitted manuscript for evaluation is an interesting research project. The combination of the external load indicator, which are the indicators recorded by the GPS system, with the internal load indicator - CK, has not been studied so far. The term COVID19 appears in the title of the paper. The timing of the study after lockdown is very specific. During the lockdown period, there was a suspension of league play. There was also a reduction in training activity. One would therefore expect a comparative compilation of the characteristics of the relationship between external and internal load before lockdown and after the start of post-lockdown games. Of course, no one planned lockdown as a variable that modifies training. That is, it would be necessary to juxtapose the correlations of the period immediately after the lockdown with the values, for example, 6-8 months later.
This is a very good point for analysis and improvement of the study. However, it was not possible to continue with this study 6-8 months later. We did the analysis in the final 10 rounds of the league. Once the league was over, the vast majority of players left for other teams (>75%) and the squad was completely new in the following season. This made it impossible to continue with the study afterwards.
Then it should be assumed that the level of preparation would no longer be determined by the break in training and competition. This is the first comment on the organization of the study, The second is the timing of blood collection and CK determinations. A post-exercise blood draw, even directly, cannot be considered to determine the baseline level. Subsequent measurements at 24, 48 and 72 hours after the match are subject to the values of CK changes that occurred as a result of training activities between matches.
Thank you very much for your comment. We have carried out methodologies similar to other studies, such as Freire et al. (2020) and Wang (2024). Due to the post-pandemic situation, we were unable to have real baseline data, and this is indicated in the limitations of the study.
Wang, S. (2024). Influence of Branched-Chain Amino Acid Ingestion on Creatine Kinase Post of Eccentric Exercise on Recovery: A Systematic Review and Meta-Analysis. Brazilian Archives of Biology and Technology, 67, e24220879.
Freire, L. D. A., Tannure, M., Gonçalves, D., Aedo-Muñoz, E., Perez, D. I. V., Brito, C. J., & Miarka, B. (2020). Correlation between creatine kinase and match load in soccer: A case report. Journal of Physical Education and Sport, 20(3), 1279-1283.
In addition to these two comments relating to the organization of the study and thus the quality of the material collected, the following points should be noted:
- the lack of CK profiles of athletes from before the experimental period. CK is a highly individualized indicator and reference values are highly variable. With individual profiles, it would be possible to operate the difference and thus standardize within the group the range of CK changes.
Thank you very much for your comment. This point is not the aim of the study, since we do not want to know previous values where the competitive calendar was normal. The aim is to know the levels of CK, related to the physical load, that is why the players are grouped into high, medium and low physical load during the congested period of matches.
- No differentiation of external loads into match and training loads
In Figure 1, the relative values of the physical demands in training can be observed (MD+1, MD+2 and MD+3), with respect to the match, therefore, a difference is made.
- There is no justification for the inclusion in the study group of a player who played in the match for 10 minutes. Here the difference in working time between 10 minutes and 60 or 90 minutes is highly significant for CK values;
Thanks for this comment. We addes the explanation in statistical analysis section.
- explain the %Δ index, i.e. the formula for determining once the essence of interpreting the value. This is especially important when the standard deviation exceeds the mean value.
This was a typographical error. We deleted it. Sorry!
- it is difficult to find an interpretation for CK changes without specific amounts of training loads in the 72 hours after the match.
In Figure 1, the relative values of the physical demands in training can be observed (MD+1, MD+2 and MD+3), with respect to the match, therefore, a difference is made.
- strength training stimulates muscle damage and CK increase, whether it was applied at 72 hours after the match;
Yes, strength training has been shown to increase CK levels, but this study was not intended to look at differences with strength training.
- no clear evidence of a relationship between the magnitude of the internal load and changes in CK values
The table shows the differences between CK levels and the load groups analysed. The results show differences between groups, not at all levels, but that is precisely why the study was carried out, to check whether or not there are differences. We cannot demand that there be evidence or clear differences in all variables.
- conclusion is very general. It does not add any new information to the topic of the manuscript. This section is unacceptable in its entirety.
The conclusion has been improved by adapting it to your indications.
The manuscript needs very extensive editorial revisions. The possibility of more readable illustrations and tables should also be studied. The current level of the text of the work does not allow to express a positive opinion.-
The manuscript has been translated by a translation company to improve its quality, as well as the clarity of the figure and table.
None of my comments caused any reflection on the part of the authors of the manuscript. I believe that disqualifying errors in the work have not been corrected. Thus, the final evaluation is negative. Research carried out in training practice cannot be deprived of methodological correctness. With such an analysis of the collected material as in the manuscript, it is difficult to call the effect of the analysis and conclusions reliable.
We are very sorry. We uploaded another document. This was a mistake. We regret the error and hope that you can help us with the new revision of this manuscript.
